# Year-Long Microbial Succession on Microplastics in Wastewater: Chaotic Dynamics Outweigh Preferential Growth

**DOI:** 10.3390/microorganisms10091775

**Published:** 2022-09-02

**Authors:** Alexander S. Tagg, Theodor Sperlea, Matthias Labrenz, Jesse P. Harrison, Jesús J. Ojeda, Melanie Sapp

**Affiliations:** 1Leibniz-Institut für Ostseeforschung Warnemünde, Seestraße 15, 18119 Rostock, Germany; 2Department of Chemical Engineering, Faculty of Science and Engineering, Swansea University, Swansea SA1 8EN, UK; 3CSC—IT Center for Science Ltd., P.O. Box 405, FI-02101 Espoo, Finland; 4Institute of Human Genetics, University Hospital Düsseldorf, Heinrich Heine University, Moorenstrasse 5, 40225 Düsseldorf, Germany

**Keywords:** microplastics, wastewater treatment, biofilm, plastisphere, chaotic dynamics

## Abstract

Microplastics are a globally-ubiquitous aquatic pollutant and have been heavily studied over the last decade. Of particular interest are the interactions between microplastics and microorganisms, especially the pursuit to discover a plastic-specific biome, the so-called plastisphere. To follow this up, a year-long microcosm experimental setup was deployed to expose five different microplastic types (and silica beads control) to activated aerobic wastewater in controlled conditions, with microbial communities being measured four times over the course of the year using 16S rDNA (bacterial) and ITS (fungal) amplicon sequencing. The biofilm community shows no evidence of a specific plastisphere, even after a year of incubation. Indeed, the microbial communities (particularly bacterial) show a clear trend of increasing dissimilarity between plastic types as time increases. Despite little evidence for a plastic-specific community, there was a slight grouping observed for polyolefins (PE and PP) in 6–12-month biofilms. Additionally, an OTU assigned to the genus *Devosia* was identified on many plastics, increasing over time while showing no growth on silicate (natural particle) controls, suggesting this could be either a slow-growing plastic-specific taxon or a symbiont to such. Both substrate-associated findings were only possible to observe in samples incubated for 6–12 months, which highlights the importance of studying long-term microbial community dynamics on plastic surfaces.

## 1. Introduction

Plastic pollution is an environmental issue with no real end in sight. Despite well-established concerns over plastic use and social pushes towards reduction of plastic use, between 4.8–12.7 Tg (teragrams) were estimated to reach global oceans in 2010 alone [1] while production proceeds annually at well over 300 Tg and is still increasing (i.e., 367 Tg in 2020, up 0.3% from 2019 [2]). Microplastics (MP; plastics < 5 mm) are a particularly important subset of this problem due to the implications MP may have for aquatic biota ingestion and food web bioaccumulation [3,4], as well as vector transport of chemicals [5,6] or microorganisms [7] due to high motility of small-sized, low-density particles. It is also worth considering that, given the manner of plastic degradation, larger plastics break down into smaller plastics. It is through this process that most MPs are created [8] so that potentially all the plastic debris in the ocean will eventually become MPs. So even if the amount of plastic waste currently reaching the oceans could be curtailed, the reservoir of potential MP bound up in the tonnes of plastic already in the deep ocean is huge.

Research into MP pollution over the last decade has been extensive. MPs have been detected in samples from all over the world, from every ocean and major sea [9,10], multitudes of lakes and rivers [11,12,13,14,15], in the air of cities [16,17] but also in highly isolated terrestrial locations [18], and even polar regions [19]. MPs’ presence and fate in wastewater treatment plants (WWTPs) have been a particular focus in recent years [20,21,22,23,24], given the interest in determining the sources of MP entering natural aquatic systems.

Given that microplastics are a global issue, understanding the ways in which these particles interact with the environment is essential. One of the most important areas for better understanding the environmental implications of microplastics is microplastic-microbial interactions, and research investigating such is now a well-established MP sub-topic [25,26,27]. Aquatic microorganisms quickly attach to surfaces and start growing biofilms (complex 3D structures composed of communities of bacteria, archaea, algae, and fungi held together by proteins and polysaccharides known collectively as extracellular polymeric substances (EPS)). Early interest in microplastic biofilms has been particularly focused on defining early colonization communities [28], particularly whether certain taxonomies preferentially associate with a specific plastic type(s) over other surfaces [29]. Indeed, the search for plastic-selective organisms referred to as the plastisphere [30,31,32], is a particular interest of such studies, given the implications such microbial communities may have for potential plastic (and plastic-additive) biodegradation. However, the term is used to describe all microorganisms attached to such particles regardless of specificity, uniqueness, or opportunistic attachment to such [33].

It has been demonstrated across a variety of different studies that biofilm communities are far more linked to other factors, such as location or habitat, than by the composition of the substrate [25,34]. However, while geographical or temporal factors typically overshadow differences between MPs and natural particles, especially between MP polymer types, some slight differences have also been reported [35,36], often in relation to more nutrient-depleted environments such as the open ocean [37]. Nevertheless, evidence for a specific plastisphere community is still absent [25], especially concerning more nutrient-rich environments. However, this could also be due to the fact that actual long-term experiments that observe the development of microbial communities in biofilms are very rare for this kind of habitats, and many studies lack suitable controls to separate opportunistic from specific attachment.

As proof of a principle for a nutrient-rich environment, aerobic activated wastewater treatment samples were examined for an entire year. The aim was to determine if, in such a distinct environment, and after longer periods of observation than has previously been applied, any clear plastic or polymer-specific biome emerges.

## 2. Materials and Methods

### 2.1. Experimental Setup

In this experiment, 10 L wastewater was collected from Severn Trent Water treatment facility in Derby, UK, and stored in a 10 L LDPE container (Thermo Scientific, Oxford, UK) stored at 22 °C and kept aerated. Air lines were kept sterile using 0.2 μm cellulose-acetate membrane syringes (VWR, Leicestershire, UK). The microcosms were set up by adding 45 mL of wastewater and 0.1 g MPs to 50 mL sterile polypropylene centrifuge tubes (VWR, Leicestershire, UK) under laminar flow. Polymer types selected for this study were polyethylene (PE), polypropylene (PP), polyvinyl chloride (PVC), nylon (PA), and polyethylene terephthalate (PET), as well as silica beads as non-plastic control. All were 150–250 μm-sized microbeads except PET, which were 1 mm-sized fragments. For sources and specifics of MPs, see Appendix A. Each tube was individually sterile-aerated, as described above. Four different time points of 1, 3, 6, and 12 months were selected to examine temporal change. Individual tubes were prepared for each polymer type and exposure period combination in triplicate (total of 72 tubes), so there were, i.e., 3 individual tubes with PE to be removed and sampled after 1 month, 3 with PE to be sampled after 3 months, etc. During exposure, direct sunlight was prevented with ambient light kept consistent to follow a diurnal cycle, the air flow rate was consistent, and ambient temperature was kept at 22 °C. Following the pre-defined exposure period, MPs were extracted from the wastewater tube, rinsed, and stored in UHQ water at 4 °C until DNA extraction. For more information on MP extraction from wastewater, see Appendix A.

### 2.2. Extraction and Sequencing

A Powersoil DNA extraction kit (MoBio, Carlsbad, CA, USA; now Qiagen, Germantown, MD, USA) was used to extract the DNA according to the manufacturer’s instructions. Due to low levels of recovered DNA, all extractions were finally eluted in 10 μL molecular grade water. Extraction blanks were included to cover each individual extraction kit used [38]. Both bacterial and fungal biofilm assemblages were studied. The V4 region of the 16S rRNA gene was targeted for investigating bacterial communities, using the primers 515F (5′-GTG CCA GCM GCC GCG GTA A-3′) and 806R (5′-GGA CTA CHV GGG TWT CTA AT-3′) [39]. For investigating plastic-associated fungal communities, the target region was ITS (Internal Transcribed Spacer) using the primers ITS-F_KYO2 (5′ TAG AGG AAG TAA AAG TCG TAA 3′) and ITS2_KYO2 (5′-TTY RCT RCG TTC TTC ATC-3′) [40]. All primers were amended with Illumina overhang adapters (Illumina’s 16S Metagenomic Sequencing Library Preparation protocol). Amplification mixes of 25 µL were carried out using the Phusion High Fidelity PCR kit (New England Bio Labs, Ipswich, MA, USA). Details for amplification and library preparation can be found in the Appendix A. Sequencing was completed using an Illumina Miseq platform with a V3 kit (Illumina, Chesterford, UK) and 2 × 300 cycles. 

### 2.3. Data Analysis

Demultiplexing was performed by the Illumina protocol using default values. The sequencing data were aligned and trimmed using the QIIME software package (MacQIIME 1.9.1-20150604) [41]. Galaxy (usegalaxy.org; accessed on 1 November 2016) [42] was used to remove blank lines in the fastq files, and bin low-quality score reads. Chimera checking and OTU picking were performed using Usearch/Uparse (v. 9.0.2132) [43,44] using UTAX trained reference databases: RDP trainset 16 for 16S rDNA and Unite v7 for ITS (UTAX Algorithm-Usearch Manual, Date Accessed: 18 August 2017). All negative controls contained no viable reads.

DNA yields were very low across the sample set, something not uncommon when extracting MP biofilms [45]. In order for reasonable comparability, rarefaction was needed for both 16S (297 sequences per sample) and ITS (499 sequences per sample). Despite this low threshold for sequences, OTU coverage of the samples was still representative of true diversity (see Appendix A for rarefaction curves).

All data analyses except for the FAPROTAX analysis were performed in R (v. 4.2.1) [46], based on OTU relative abundance. Bray-Curtis distances between samples were constructed, and principle coordinate analyses were calculated using the R package ecodist (v. 2.0.9) [47]. Heatmaps were visualized using the R package ComplexHeatmap (v. 2.13.1) [48]. All other figures were created using the R package ggplot2 (v. 3.3.6) [49] with additional functionality from the R packages ggpubr (v. 0.4.0) [50] and ggforce (v. 0.3.3) [51]. The indicator species analysis was performed using the multipatt function in the R package indicspecies (v. 1.7.12) [52] on log-transformed microbial abundances with default parameters and 9999 permutations for *p*-value calculation. Only associations with a *p* value lower than 0.05 were analyzed further. Functional profiles of the samples were derived from the microbial community composition data using the script collapse_table.py from the FAPROTAX software suite (v. 1.2.4) [53]. Statistical analysis of similarities (ANOSIM) between groups was also completed in R using the “vegan” library (permutations = 999; distance = “bray”) [54]. Alpha diversity boxplots (both OUT richness and Shannon-Weiner) are included in the Appendix A (see Appendix A).

16S and ITS raw sequences have been deposited in the National Centre for Biotechnology Information (NCBI) Sequence Read Archive (SRA) under BioProject accession ID PRJNA861953.

## 3. Results

A total of 148 OTUs were identified across the rarefied dataset (87 16S and 61 ITS). 16S results showed a clear succession pattern of shifting community composition across the year, with increases in dissimilarity as the year progressed (Figure 1).

Polymer type was not found to have any clear influence on composition across the year, confirmed by ANOSIM analysis (R = 0.04453, *p* = 0.133). For both 16S and ITS, the highest similarity was seen in 1-month samples and greatly increased in dissimilarity following this point (Figure 1 and Appendix A). As such, the length of incubation had a strong effect on determining biofilm similarity, while substrate features (physiology/chemistry) had no marked influence on succession patterns.

Figure 2 demonstrates the grouping observed according to incubation time along with the prevalence of OTUs driving these distinctions. It can be observed that while 1 and 3-month samples form clearly distinct groupings, 6- and 12-month samples do not. Combined with the results of Figure 1, this indicates that while dissimilarity increases throughout the year, the changes after six months appear not distinct enough to consider 6- and 12-month communities as fundamentally different. The clades (A–C) in the heatmap give indications as to how these sample groupings of 1 month, 3 months, and 6–12 months are defined. The 1-month samples are most clearly defined by sharing very low coverage in clade C, totally lacking 4 OTUs, which appear prominently in both other groups. The 6–12 months group appears to be clearly defined by low coverage in clade B, combined with better coverage in C and weaker coverage in A, marked particularly by near total lack of *Corynebacterineae* (OTU 11), an OTU well-represented in the other groups. The 3-month samples appear to be distinguished by having reasonable coverage across all clades. 

When functional activity in the communities is estimated using FARPROTAX, an overall trend can be seen in reducing functional diversity over time (see Figure 3). Some key functions of wastewater biofilms appear to be always present, namely chemoheterotrophy and aerobic chemoheterotrophy; however, other functions see a marked decrease following three months.

Indicator analyses were performed to determine which taxa specifically only appear at certain time points. There are two ways an OTU can be indicative for a given time point; either by the significant presence in the given time point and significant absence in others or by the significant presence in all other time points with significant absence in the given time point. The. 1-month samples have the most indicators, with a total of 14 indicative OTUs (7 by presence and 5 by absence). The most significant of these was *Chitinophaga* sp. (OTU 28). 6 OTUs were identified as indicators for 3-month samples (5 by presence and 1 by absence), and the most significant of these was *Romboutsia* sp. (OTU 53). While six indicators were identified for 6-month samples and 1 for 12 months, given the prior results demonstrating that these two time points can generally be considered as a single group of >6 months (see, e.g., Figure 2), it also follows that indicators for this collective time point are more useful. In total, 11 OTUs were indicative of the >6 month time point (all by presence), and the most significant was an unassigned taxon in the order *Terriglobus* (OTU 29).

Differences specific to polymer type are difficult to determine, given the strength of the effect of exposure time. Indicator analysis performed for substrate type did, however find some significant indicators for certain plastics. No substrate-specific OTUs were found for PP. PVC had both a potential bacterial indicator (*Sulfuriferula* sp.; OTU 59) and a fungal (unassigned taxon in the class Agaricomycetes; OTU 48); for PET this was the bacterium *Chthonomonas* sp. (OTU 58), for PE the bacterium *Mucilaginibacter* sp. (OTU 380) and for PA the fungus *Malassezia* sp. (OTU 24). Indicator analysis also revealed one OTU, the Alphaproteobacterium *Devosia* sp., which appeared on all MP substrates but not on the silica control.

## 4. Discussion

Polymer type appeared to have a minor effect on the studied microbial community across the year and shows instead an example (especially for the 16S rRNA gene sequencing data) of succession best defined by chaos theory. Given that external factors are controlled for or kept consistent throughout, it might be expected that, in the absence of a polymer-type effect (the variable under study), biofilm succession should be the same for every sample throughout. However, this is only likely true if starting setups are truly identical. Since this is not the case (and is practically impossible), the starting points are all very subtly different, and the colonizing biofilms, while likely highly similar, will not be completely identical [55,56]. Chaos theory describes systems in which infinitesimal differences in initial conditions become amplified over time [57], and has been demonstrated, either directly in microbial systems [58,59] or used to predict microbial community dynamics [60]. Thus, we would expect to see samples most similar early on, with differences increasing over time. This is exactly what is observed in this study and particularly well-demonstrated in the 16S rRNA gene data. This might explain why polymer type has little to no influence on overall biofilm succession. 

Even when controlling for the known influencing factor of incubation time, no clear grouping of polymer types or even plastics compared with silicate substrates could be identified. This is particularly visible in Figure 2, where no group of 3 substrate replicates cluster together, even within the already-defined incubation time-associated groups. However, this is not necessarily unexpected. Indeed, many studies have shown no clear effect on polymer type in a variety of other environments [61,62,63], and a previous study from within a WWTP (albeit testing other stages of treatment than the aerobic activated) also showed little difference between plastics and silica bead controls [64]. However, Figure 2 does show one cluster, which, if some definitions are widened, could be indicative of a substrate chemistry-associated influence. As applied prior, the 6- and 12-month samples can be combined to be thought of as a >6 months grouping. Then within this, PE and PP can be combined and considered collectively as polyolefins, and a clustering can be observed (demarked in Figure 2 by the red box). This cluster is particularly consistently represented by clade C OTUs, especially *Dokdonella*, *Corynebacterineae*, *Terriglobus,* and Pseudonocardineae (OTUs 15, 19, 29, and 38, respectively). However, there is no evidence from the literature that any known strains of *Dokdonella* or *Terriglobus* have plastic-degrading potential. Two species of *Dokdonella* have been identified in activated, aerobic wastewater. However, no growth has been reported on longer chain hydrocarbon media [65,66,67]. *Terriglobus* has five species in the genus, but none have thus far been linked to plastic or even petrochemical oil metabolism [68,69]. As for the two unassigned taxa from the orders *Corynebacterineae* and *Pseudonocardineae*, without more definition, any such speculation is trivial. Therefore, NCBI BLAST was used to try and provide a better definition. The best match for OTU 19 (*Corynebacterineae*) was *Mycobacterium fortuitum* [70], which was isolated from Iranian hospital water, and for OTU 38 (*Pseudonocardineae*) was *Pseudonocardia xishanensis* [71] which was found in sterilized wormwood (*Artemisia annua*) root. Again, neither have any recorded history of petrochemical degradation. So, it is unclear why these taxa appear to be more selectively appearing in polyolefin biofilms after six months of incubation in activated wastewater. Still, it may be interesting for future research into these topics.

However, one taxon which is particularly interesting from a plastisphere (or plastic-specificity) perspective is *Devosia*. This genus appears in many different MP biofilms over the year but, crucially, does not appear on any of the silicate (natural particle) controls. When examined in more detail, the presence of this OTU appears to be increasing slowly over the course of the year. Figure 4 demonstrates this trend. While this is just a fraction of the biofilm (average relative abundance 0.11% at 1 month growing to 0.44% at 12 months, it is nevertheless interesting that this OTU never developed on any of the silica samples. However, literature on the genus is limited. BLAST results revealed no further information with regard to possible species, so the reasons behind why this taxon was only found on plastic (indeed on 85% of PVC samples and 75% of PA) and not silica is unclear. *Devosia* strains are involved in the degradation of the plant toxin deoxynivalenol [72,73], while one study showed bacterial strains identified as potentially *Devosia* sp. with the ability to degrade dibenzothiophene, an organosulphur compound found in petroleum [74]. However, while it is not possible to say whether *Devosia* is in some way concerned with plastic metabolization based on these findings alone, these results do merit further study, especially given the interest in discovering potential plastic degraders. Future studies should include other methods to detect and quantify members of this genus within the plastisphere to overcome the limitations associated with 16S rDNA sequencing. 

Given these developments in the bacterial biofilm communities across the year, with clear shifts in composition occurring, it is interesting to see the accompanying shifts in functional dynamics of decreasing functional diversity after three months of incubation. Moreover, of particular interest is that the human pathogens are almost completely absent at one month but at three months are well distributed across the samples, before disappearing again at six months and later. It is not clear why this may be. Still, growth in other functions appears to suggest that this could be linked to more fundamental shifts in metabolism dynamics, given that phototrophy and photoheterotrophy also follow a very similar shift to that seen for human pathogens. However, these functional links to 16S rRNA gene sequencing data made by FAPROTAX only serve as a very rough estimate of functional activity. As such, these observations should only be considered of interest for possible further research rather than a clear result of functional changes. Future studies should take into account a more functional view by applying metagenomic or metatranscriptomic analyses.

Thus far, this discussion has been centered on the bacterial biofilm communities. While fungal biofilms demonstrated this same pattern of chaotic succession up to 6 months as in 16S rDNA, the 12-month fungal data shows an, albeit small (6% average similarity), increase in similarity between 6–12 months, indicating that this chaotic drift seen in samples, at least regarding fungal communities, reaches an endpoint. Interestingly, these findings are generally similar to those by Wallbank et al. [75], who also examined MP bacterial and fungal biofilm succession over time, albeit in seawater between 2 and 12 weeks. They demonstrated the highest similarity at the first time point, comparable for both bacteria and fungi, and also saw complete shifts in bacterial communities over time while seeing less fundamental shifts with more overlap in fungal communities, all of which is similar to what was observed within our study. However, they did not observe the step-wise increase in dissimilarity seen in this study for bacteria but instead saw the highest dissimilarity at 6 weeks and reduced dissimilarity at 12 weeks. It must be noted, however, that their study is environmental, and as discussed earlier, changes in environmental parameters are likely to cause changes in biofilm communities. Nevertheless, it is clear that fungal succession does not occur in the same way as bacteria, and given that certain fungi have also been demonstrated to be plastic degraders [76,77], it highlights the importance of including fungi when studying microplastic biofilm dynamics. Furthermore, future mesocosm studies need to take into account changes in chemical parameters, including nutrients and potential substrates such as carbohydrates, to better link observed shifts to the biochemistry in the closed system studied.

## 5. Conclusions

Incubation of MPs over the course of a year in controlled wastewater microcosms revealed that MP biofilms become more and more differentiated over time, with no clear evidence of a specific influence from differing polymer types. As such, there is little evidence of a specific plastisphere on MPs in wastewater. MPs in wastewater appear to be colonized in a similar manner to similarly-sized natural particles. However, after six months a slight differentiation between polyolefins and other substrates was observed. The occurrence and increase of growth of *Devosia* in MP biofilms (while completely absent in natural particle [silicate] controls) over the course of the year may be indicative of a plastic-specialized taxon, but further studies are needed to verify this specialization. In general, however, these findings highlight the importance of longer-term, especially >6 months, studies into MP biofilms, as both findings were only observable in 6–12 month samples 6.

## Figures and Tables

**Figure 1 microorganisms-10-01775-f001:**
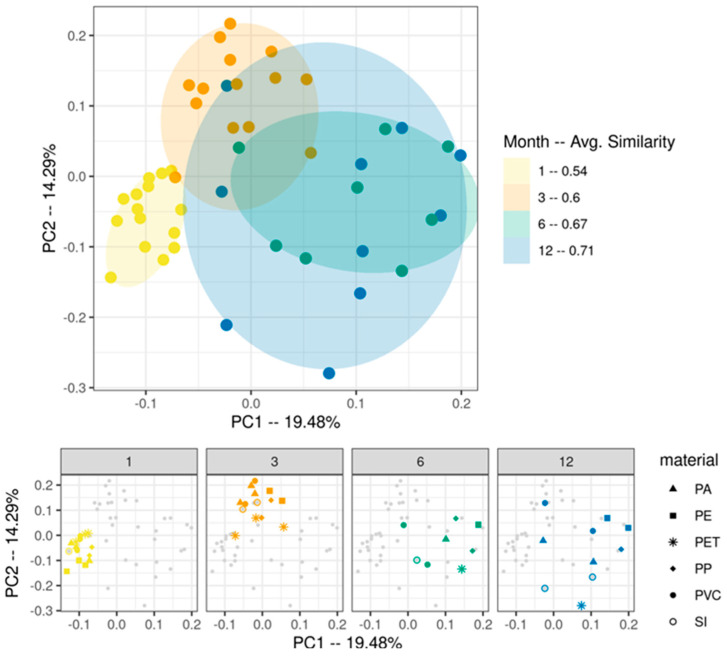
Principle coordinate analyses of 16S data based on Bray-Curtis dissimilarity matrix. The top plot shows the overall dataset, while the bottom plots give details on polymer types. PA: polyamide; PE: polyethylene; PET: polyethylene terephthalate; PP: polypropylene; PVC: polyvinyl chloride; SI: silica (control).

**Figure 2 microorganisms-10-01775-f002:**
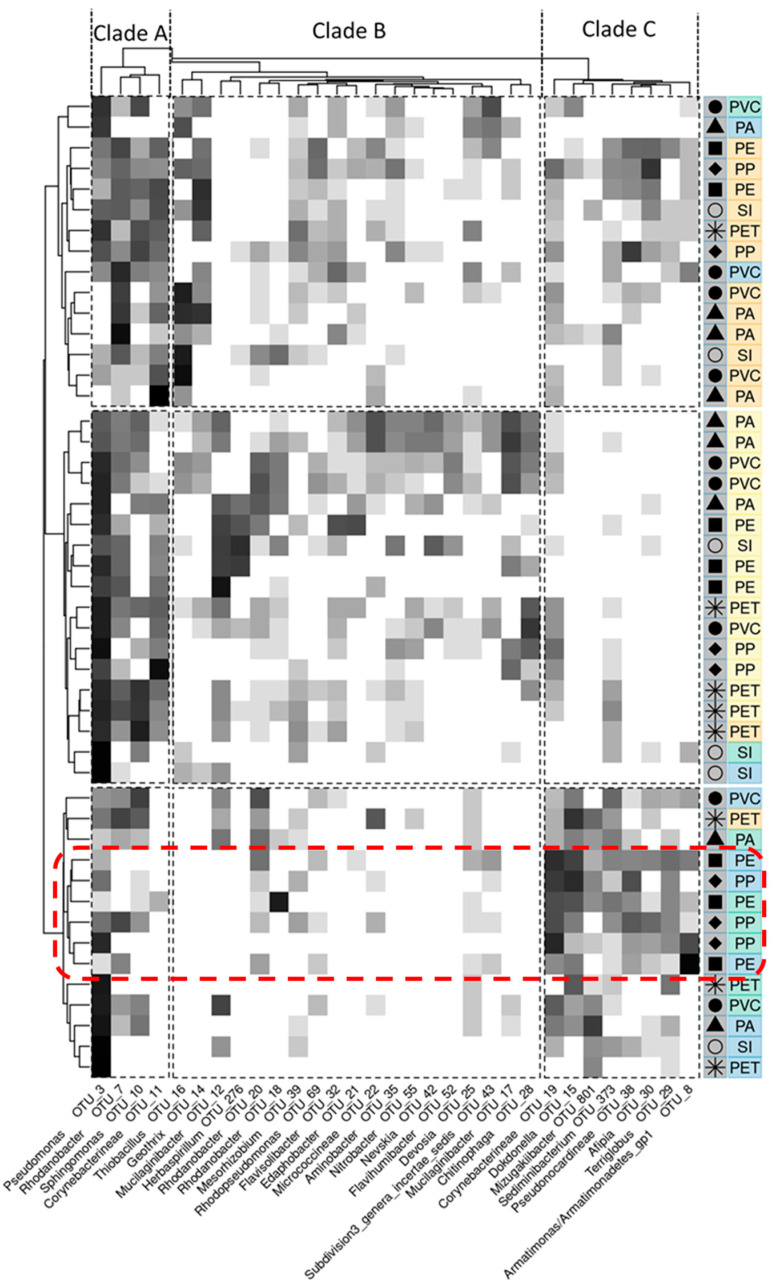
Heatmap based on log-transformed microbial abundances. Colors and icons refer to exposure time and substrate type, as shown in Figure 1. Groups of samples and clades of OTUs are demarked by black dashed lines and defined based on k-means clustering of the euclidean distance between samples. Dendrograms represent euclidean distance clustered via complete linkage. Red box refers to a possible sub-group of > 6-month polyolefins. PA: polyamide; PE: polyethylene; PET: polyethylene terephthalate; PP: polypropylene; PVC: polyvinyl chloride; SI: silica (control).

**Figure 3 microorganisms-10-01775-f003:**
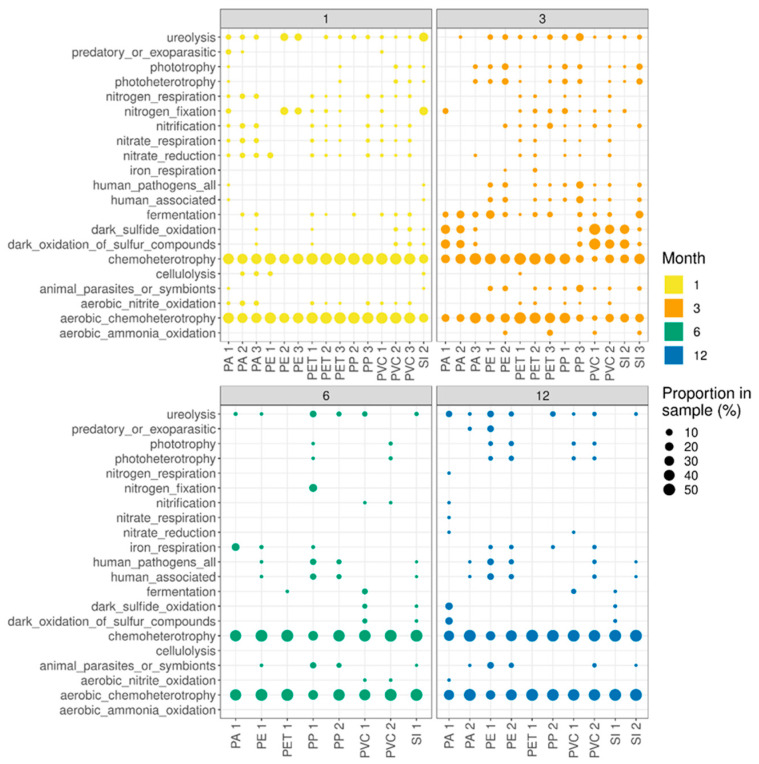
Inferred functional diversity based on FAPROTAX. PA: polyamide; PE: polyethylene; PET: polyethylene terephthalate; PP: polypropylene; PVC: polyvinyl chloride; SI: silica (control).

**Figure 4 microorganisms-10-01775-f004:**
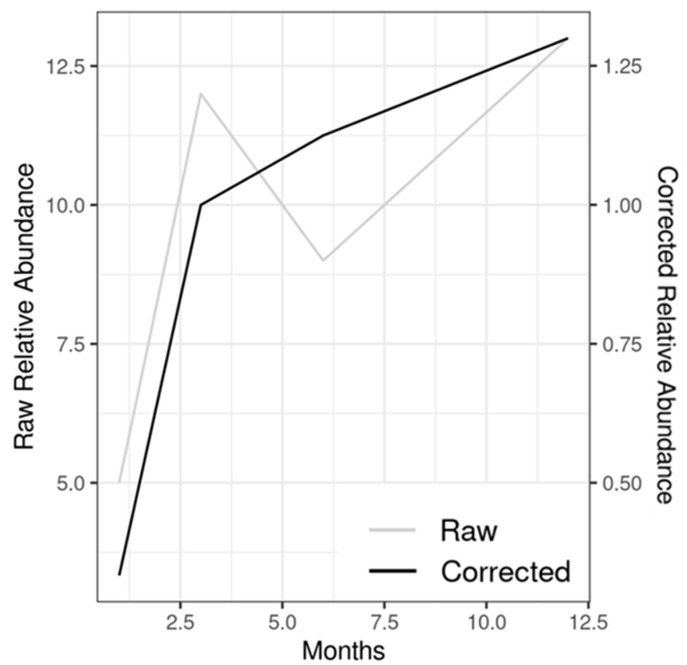
Read counts across the entire dataset of *Devosia* (OTU 25). Raw refers to the total number of reads in a rarefied dataset per time point, while the corrected refers to read count averaged by a number of samples per time point. The reduction in raw count at six months (despite the average gain) is due to the relatively greater loss of 6-month samples due to them not reaching rarefication threshold.

## Data Availability

The data presented in this study are openly available from the NCBI Sequence Read Archive (SRA) under the BioProject PRJNA861953.

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
