# Peer review of "Year-Long Microbial Succession on Microplastics in Wastewater: Chaotic Dynamics Outweigh Preferential Growth"

_microorganisms, 2022, doi:10.3390/microorganisms10091775_

Round 1

Reviewer 1 Report

I found this article interesting and indeed highlighting a knowledge gap of importance. I have reviewed the manuscript,

Tilted; Ingested Microplastics In Local Fish Species From The NW Mediterranean Sea By Mel Constant * , Mathieu Reynaud , Lisa Weiss , Wolfgang Ludwig , Philippe Kerhervé

Does the paper present novel ideas/a novel direction with regard to the field of research? ; Is the research rationale sound? ; Is the literature review complete and which other papers can the author cite?; Are the research implications clearly mentioned?; If they are mentioned, are they sound?; Are the concluding statements clear?; Is the research design appropriate? What are the gaps, and what should be done to fill the gaps?; Is the research methodology sound and relevant to the field?; Do the main ideas in the paper flow well? Was the flow of ideas/the main argument natural?

After reviewing everything, I think the paper deals with interesting subject and is worthy of publication, I think that this paper could help researchers who would like to make a study of the aquatic environment of microplastics with microorganisms.

Author Response

We thank the reviewer for taking the time to review the manuscript and for the positive endorsement for publication. 

Reviewer 2 Report

Revised manuscript studies microplastics problems which new and very serious threat to the environment as nowadays one can find it in all environmental compartments. In addition, there is hardly any knowledge of microorganisms formic biomes connected with plastics. This topic is important to study as knowing any microbes associated with plasticsphere would give some potential microbes for plastic degradation.

Taking into account all above I find the manuscript falling into the scope of Microorganisms Journal as well as important topic for publication.

The title is good, keywords appropriate and abstract provides general description of the manuscript content. The language is good but requires some corrections as there are many grammar issues.

Introduction presents up to date knowledge about microplastics with good literature giving essential information. It ends with a clear aim.

I have question about control - in abstract it was silicon beads while in methods it was glass beads. Please correct this. In addition, there is no abbreviation for this treatment. On figures one can see SI - is it control? Again does it stands for silicon beads?

I wonder if there could be any influences of container material (LDPE, centrifuge tubes - what kind material were they made?) on plastic levels and microbial growth. Could you comment on this?

Besides this experiment was well designed and conducted. Molecular and bioinformatic part is very good.

Results are clearly described with high quality figures supported by supplementary materials. This gives the feeling of good study and the results can be trusted. They are also quite well discussed leading to conclusions.

Line 317 - please write Devosia in italics.

It was very first work on plasticsphere so Authors did good job to start this new topic. For that reason the manuscript is worth to be published after tiny corrections.

Author Response

Revised manuscript studies microplastics problems which new and very serious threat to the environment as nowadays one can find it in all environmental compartments. In addition, there is hardly any knowledge of microorganisms formic biomes connected with plastics. This topic is important to study as knowing any microbes associated with plasticsphere would give some potential microbes for plastic degradation.

Taking into account all above I find the manuscript falling into the scope of Microorganisms Journal as well as important topic for publication.

The title is good, keywords appropriate and abstract provides general description of the manuscript content. The language is good but requires some corrections as there are many grammar issues.

Introduction presents up to date knowledge about microplastics with good literature giving essential information. It ends with a clear aim.

We thank the reviewer for taking the time to review our work so extensively. We are pleased the reviewer recognised the importance of the topic and novelty of the work. 

I have question about control - in abstract it was silicon beads while in methods it was glass beads. Please correct this. In addition, there is no abbreviation for this treatment. On figures one can see SI - is it control? Again does it stands for silicon beads?

We have changed any mention of glass beads to silica and additionally clarified the abbreviations for substrates (including for silicon bead control) in all relevant figure legends

I wonder if there could be any influences of container material (LDPE, centrifuge tubes - what kind material were they made?) on plastic levels and microbial growth. Could you comment on this?

The material (polypropylene) and manufacturer of the centrifuge tubes (VWR, Leicestershire, UK) has been added to the text. It is true the material of the tubes and transport bottles may have an effect on the community, although this is true for the microcosm experimental setup as a whole, and it is practically certain that the communities in the microcosm differ to those in the environment because of the experimental setup, in something known as “the bottle effect”, which is a well-established phenomenon in microbial research. However, this is not considered problematic providing the setup is consistent across all the samples insofar as possible, and in the case of the materials of the laboratory equipment used, this was indeed consistent across all samples and therefore would not be responsible for any differences observed between samples. Nevertheless, such an issue has been highlighted as a limitation of such setups in the manuscript, along with where future research might improve (see lines 332-5).

Besides this experiment was well designed and conducted. Molecular and bioinformatic part is very good.
Results are clearly described with high quality figures supported by supplementary materials. This gives the feeling of good study and the results can be trusted. They are also quite well discussed leading to conclusions.

We thank the reviewer for the positive endorsement of our experimental work

Line 317 - please write Devosia in italics.

The text has been updated with the requested change

It was very first work on plasticsphere so Authors did good job to start this new topic. For that reason the manuscript is worth to be published after tiny corrections.

We thank the reviewer for their many kind comments and appropriate suggestions for additional clarification and believe the paper is now much improved. 

Reviewer 3 Report

The manuscript “microorganisms-1886676” mainly develop “Year-Long Microbial Succession on Microplastics in Wastewater: Assessing Chaotic Dynamics Vs Preferential Growth”. In my opinion, the experimental design is relatively reasonable and innovative, and the characteristics of the development process are sufficient, but it may be recommended to consider acceptance after major revisions.

1.    The title of paper is assessing chaotic dynamics vs preferential growth, but there is no comparison in the text. It is recommended to replace.

2.    In this manuscript, the author thinks “Four different time points of 1, 3, 6 and 12-months were selected to examine temporal change.” (p3, line92-93). How does the author demonstrate these four time points in time have different meaning? Please supplement this question.

3.    “Individual tubes were prepared for each polymer type and exposure period combination in triplicate (total of 72 tubes).” (p3, line93-94). The same sample is sampled independently in each time period. How to ensure the reproducibility of the results in this paper. Please supplement this question.

4.    The line of raw in Figure 4. How to explain that the raw relative abundance first increases, then decreases and finally increases with the change of time?

5.    The novelty of this research should be well presented in the Introduction. The introduction could be written better. Several relative papers are suggested to be cited. (10.1016/j.jhazmat.2022.12859710.1080/10408398.2022.2105801).

6.    The amount of data in the article is insufficient, consider whether to add other relevant data.

7.    The paper have errors in the reference, please check carefully. For example,19. Kanhai, L. D. K., Gardfeldt, K., Krumpen, T., Thompson, R. C. & O’Connor, I. Microplastics in sea ice and seawater beneath ice floes from the Arctic Ocean. Sci. Rep. 10, 1–11 (2020).” There are five authors listed.

Author Response

The manuscript “microorganisms-1886676” mainly develop “Year-Long Microbial Succession on Microplastics in Wastewater: Assessing Chaotic Dynamics Vs Preferential Growth”. In my opinion, the experimental design is relatively reasonable and innovative, and the characteristics of the development process are sufficient, but it may be recommended to consider acceptance after major revisions.

1.    The title of paper is assessing chaotic dynamics vs preferential growth, but there is no comparison in the text. It is recommended to replace.

The title has been updated accordingly to remove the suggestion this was an active comparison rather than an observation in the results. We feel the new title is much more appropriate and thank the reviewer for pointing this out.

2.    In this manuscript, the author thinks “Four different time points of 1, 3, 6 and 12-months were selected to examine temporal change.” (p3, line92-93). How does the author demonstrate these four time points in time have different meaning? Please supplement this question.

Over time we would expect to see transitional succession from early colonisers to short-term established biota, eventually transitioning to long-term (and possibly ideally-adapted or specialised) persistent members of the community. It is not known when such transitions are likely to take place and indeed how long it might take for the long-term specialists to become truly established to the point of detection using NGS, and indeed this lack of knowledge about longer term (>6 month) succession is presented in the introduction as a point of novelty about the research (see lines 75-78). Interval selection was made to best catch such successional changes. This is why intervals were more frequent at the earlier stages of the experiment, as we would expect more dynamic changes to occur earlier as colonisers give way to more established members, while changes in the more stable, established biofilm were expected to take longer, hence larger intervals.

3.    “Individual tubes were prepared for each polymer type and exposure period combination in triplicate (total of 72 tubes).” (p3, line93-94). The same sample is sampled independently in each time period. How to ensure the reproducibility of the results in this paper. Please supplement this question.

The text has been updated to clarify this setup. Line 98-99 now has the addendum:
“.. in triplicate (total of 72 tubes), so there were i.e. 3 individual tubes with PE to be removed and sampled after 1 month, 3 with PE to be sampled after 3 months etc.”

4.    The line of raw in Figure 4. How to explain that the raw relative abundance first increases, then decreases and finally increases with the change of time?

The reason the “raw” (un-corrected) count of Devosia sp. decreases while the corrected line shows an increase is because at 6 months many samples were lost due to not reaching the rarefication threshold. Therefore, the total count was lower at 6 months than at 3 months, however the number of samples at 3 months was much higher, and when the count is averaged for the number of samples, this results in the small increase observed in the “corrected” line. This clarification has been added to the figure legend (lines 285-7):
“The reduction in raw count at 6 months (despite the average gain) is due to the relatively greater loss of 6 month samples due to them not reaching rarefication threshold.”

5.    The novelty of this research should be well presented in the Introduction. The introduction could be written better. Several relative papers are suggested to be cited. (10.1016/j.jhazmat.2022.128597,10.1080/10408398.2022.2105801).

Thank you for the suggestion. These papers concern carbon nanomaterials and carbon-dot therapeutics and as such we do not believe they are relevant.

6.    The amount of data in the article is insufficient, consider whether to add other relevant data.

Thank you for the suggestion. Additional Basic local alignment tool (BLAST) matches for all OTUs can be found in the supplementary materials. While these do provide additional data, given the size of the tables we believe they are better placed in the supplementary materials than the main text. 

7.    The paper have errors in the reference, please check carefully. For example, “19. Kanhai, L. D. K., Gardfeldt, K., Krumpen, T., Thompson, R. C. & O’Connor, I. Microplastics in sea ice and seawater beneath ice floes from the Arctic Ocean. Sci. Rep. 10, 1–11 (2020).” There are five authors listed.

We have re-checked the reference list for any errors. 

Round 2

Reviewer 3 Report

ok